# Objective and data-driven Bayesian inference using TabPFN models

**Elias Chaibub Neto** [1]

## Abstract

Neural posterior estimation with prior-data fitted networks (NPE-PFN) has emerged as a promising tool for Bayesian inference, enabling posterior approximation from simulated data generated under user-specified prior and likelihood models. Here, we adapt TabPFN-based NPE-PFN modeling for performing both objective Bayesian inference and data-driven Bayesian analysis. Importantly, while both contributions can be strait-forwardly implemented under the NPE-PFN framework, they cannot be implemented (or at least not as easily) under alternative computational approaches. In the objective Bayes front, we propose a new class of default prior distributions for which maximum a posteriori (MAP) and maximum likelihood (MLE) inferences coincide. These priors yield MAP/MLE equivalence independent of the chosen probability model, enabling more automatic Bayesian analysis without model-specific prior design. On the data-driven front, we show how the NPE-PFN machinery can be used to approximate MLE estimation through overconfident asymptotic Bayesian arguments allowing the implementation of empirical Bayes methodology. Simulation studies illustrate the flexibility and effectiveness of the proposed approaches.

## 1. Introduction

Tabular foundation models (TFMs) have recently been adapted to perform Bayesian inference in the context of simulation-based inference (Vetter et al., 2025). In this paper, we describe how to use TFMs for performing objective and data-driven Bayesian inference with standard parametric probability models widely used in statistics.

Following Vetter et al. (2025), we adopt the NPE-PFN approach based on the TabPFN model (Hollmann et al., 2023;

Hollmann et al., 2025). Its key practical advantage is that it relaxes many of the modeling constraints traditionally imposed by analytical or computational convenience. Classical Bayesian workflows often rely on conjugate priors, closed-form full conditional distributions for Gibbs sampling, or models that are otherwise compatible with existing MCMC and variational inference algorithms. By contrast, the NPE-PFN approach only requires the ability to generate samples from the prior and probability model distributions. This makes it straightforward to implement flexible Bayesian models, including higher-level hierarchical models.

This flexibility is particularly useful for objective Bayesian inference. Bayesian inference is often framed along a spectrum between subjective and objective paradigms, distinguished primarily by how prior information is integrated into the analysis. In the subjective tradition, priors represent coherent degrees of belief for an agent and are ideally informed by substantive expertise (de Finetti, 1937; Savage, 1954; Lindley, 1972). In many applications, however, substantive expertise is unavailable, difficult to formalize, or insufficient for eliciting an informative prior. This common situation has motivated the development of the objective, or "default" Bayesian school of statistics, which seeks to construct principled priors that minimize subjectivity and enable Bayesian inference when genuine prior information is limited. This tradition focuses on priors based on formal criteria that allow the likelihood to dominate the analysis while retaining Bayesian coherence (Jeffreys, 1939; Bernardo, 1979; Bernardo and Smith, 1994; Berger, 1985; Kass and Wasserman, 1996). Among these proposals, reference analysis (Bernardo, 1979; Bernardo and Smith, 1994) is the most widely adopted in applications.

Beyond classical default-prior constructions, other Bayesian approaches use data-driven priors. These procedures can also be considered objective in the sense that they reduce reliance on subjective elicitation by learning an informative prior directly from data. A classical example is empirical Bayes (Carlin and Louis, 2008; Efron, 2010).

The NPE-PFN framework provides a natural computational setting for these broader views of objective and data-driven Bayesian analysis: rather than restricting attention to priors and models chosen for analytical tractability, one can specify flexible prior-generating mechanisms and likelihood models, simulate from them, and use the trained founda-

[1]Sage Bionetworks, Seattle, Washington, United States of America. Correspondence to: Elias Chaibub Neto <elias.chaibub.neto@sagebase.org>.

*Proceedings of the $2^{nd}$ ICML Workshop on Foundation Models for Structured Data*, Seoul, South Korea. 2026. Copyright 2026 by the author(s).

tion model to approximate the resulting posterior inference. However, as described in more detail in the Related work section (Appendix A.2 and A.3), the direct application of the NPE-PFN framework to these analyses is problematic and requires the development of new criteria and methodology for objective and empirical Bayes inference.

Our contributions are twofold. First, we propose a new criterium for objective Bayesian inference which requires MAP and MLE inferences to coincide. We show that a hierarchical uniform-half-t prior satisfies this criterion for any arbitrary probability model containing parameters with unbounded support, whereas a uniform prior satisfies the criterion in the bounded parameter space case. In contrast to standard objective reference analysis, where each probability model typically requires a different default prior, our approach uses the same prior across distinct probability models, enabling more automatic statistical analysis.

Second, we describe how to perform empirical Bayes (EB) analyses under the NPE-PFN framework. To this end, we describe how to approximate MLE estimation using the NPE-PFN machinery using overconfident asymptotic Bayesian results. We also illustrate how statistical inference can be improved in difficult flat-likelihood settings by applying EB to a two-stage hierarchical Bayesian model. This example highlights a setting where the proposed EB solution is straightforward to implement within NPE-PFN but considerably more difficult to implement using traditional Bayesian inference approaches.

Related work and background on the NPE-PFN approach are presented in Appendixes A and B, respectively.

## 2. Contributions

### 2.1. Default priors satisfying MAP/MLE equivalence

Let $p(\boldsymbol{x} \mid \boldsymbol{\lambda})$ represent an arbitrary probability model parameterized by $\boldsymbol{\lambda} = \{\lambda_1, \ldots, \lambda_l\}$, where each $\lambda_j$ can be either bounded in a $[a_j, b_j]$ interval (e.g., probability of success in a bernoulli model) or unbounded assuming values in either $\mathbb{R}$ or $\mathbb{R}^+$ (e.g., mean and variance of a normal model).

Starting with the unbounded parameter space case, where $\lambda_j \in \mathbb{R}$ or $\lambda_j \in \mathbb{R}^+$, let $p(\boldsymbol{\lambda} \mid \boldsymbol{\phi}) \, p(\boldsymbol{\phi})$ represent an hierarchical prior assuming the form,

$$p(\boldsymbol{\lambda} \mid \boldsymbol{\phi})p(\boldsymbol{\phi}) = \prod_{j=1}^{l} p(\lambda_j \mid \phi_j)p(\phi_j) = \prod_{j=1}^{l} U(a, \phi_j)p(\phi_j),$$
(1)

where $a = 0$ if $\lambda_j \in \mathbb{R}^+$, $a = -\phi_j$ if $\lambda_j \in \mathbb{R}$, and $p(\phi_j)$ represents a probability distribution with support in $\mathbb{R}^+$.

Under this prior specification, the following result (proven in Appendix D.1) holds.

**Theorem 1.** *Let $\lambda_j \in \mathbb{R}$ or $\lambda_j \in \mathbb{R}^+$, for $j = 1, \ldots, l$. Then under the prior in equation 1, the MAP of the marginal*

*posterior distribution of $\boldsymbol{\lambda}$, $MAP_{p(\boldsymbol{\lambda}|\boldsymbol{x})} = \operatorname{argmax}_{\boldsymbol{\lambda}} p(\boldsymbol{\lambda} \mid \boldsymbol{x})$, equals the MLE, $\hat{\boldsymbol{\lambda}} = \operatorname{argmax}_{\boldsymbol{\lambda}} p(\boldsymbol{x} \mid \boldsymbol{\lambda})$.*

While the above result holds for any prior $p(\phi_j)$ with support on $\mathbb{R}^+$, the half-t distribution, $p(\phi_j) \sim ht_{d_{\phi_j}}(s_{\phi_j}^2)$ provides a good practical choice due to its flexibility for representing light or heavy tail distributions. Throughout the text we will denote the following hierarchical prior, $p(\boldsymbol{\lambda} \mid \boldsymbol{\phi}) \, p(\boldsymbol{\phi}) = \prod_{j=1}^{l} U(a, \phi_j) \, ht_{d_{\phi_j}}(s_{\phi_j}^2)$, as the uniform-half-t prior. Importantly, note that the same uniform-half-t prior can be used as a default prior for any arbitrary probability model $p(\boldsymbol{x} \mid \boldsymbol{\lambda})$ with unbounded parameter space.

Similarly, in the bounded parameter space case, where $\lambda_j \in [a_j, b_j]$, the analogous result based on a uniform prior (proven in Appendix D.2) holds.

**Theorem 2.** *Let $\lambda_j \in [a_j, b_j]$. Then under the prior $p(\boldsymbol{\lambda}) = \prod_{j=1}^{l} p(\lambda_j)$, where $p(\lambda_j) \sim U(a_j, b_j)$ the MAP of the posterior distribution of $\boldsymbol{\lambda}$, $MAP_{p(\boldsymbol{\lambda}|\boldsymbol{x})} = \operatorname{argmax}_{\boldsymbol{\lambda}} p(\boldsymbol{\lambda} \mid \boldsymbol{x})$, equals the MLE, $\hat{\boldsymbol{\lambda}} = \operatorname{argmax}_{\boldsymbol{\lambda}} p(\boldsymbol{x} \mid \boldsymbol{\lambda})$.*

These results show that for any arbitrary probability model parameterized by $\boldsymbol{\lambda} = \{\lambda_1, \ldots, \lambda_l\}$ we can obtain MAP/MLE equivalence inferences by adopting a $U(-\phi_j, \phi_j) \times ht_{d_{\phi_j}}(s_{\phi_j}^2)$ prior when $\lambda_j \in \mathbb{R}$, a $U(0, \phi_j) \times ht_{d_{\phi_j}}(s_{\phi_j}^2)$ prior when $\lambda_j \in \mathbb{R}^+$, and a $U(a_j, b_j)$ prior when $\lambda_j \in [a_j, b_j]$. So, essentially, our MAP/MLE equivalence prior will be composed by a combination of these 3 default choices according to the support of the $\lambda_j$ parameter.

Finally, note that the MAP/MLE equivalence criterium was motivated by difficulties to implement reference analysis using NPE-PFN (see Appendix A.2 for further details).

### 2.2. NPE-PFN-based Empirical Bayes

Empirical Bayes (EB) methods provide a principled compromise between fully Bayesian and frequentist approaches. For a Bayesian model with likelihood $p(\boldsymbol{x} \mid \boldsymbol{\lambda})$ and prior $p(\boldsymbol{\lambda} \mid \boldsymbol{\phi})$ with unknown hyperparameter $\boldsymbol{\phi}$ and where the main interest is about $\boldsymbol{\lambda}$ inferences, the fully Bayesian solution is to assign a prior $p(\boldsymbol{\phi})$ to $\boldsymbol{\phi}$, and compute the marginal posterior distribution of $\boldsymbol{\lambda}$, $p(\boldsymbol{\lambda} \mid \boldsymbol{x}) = \int_{\boldsymbol{\phi}} p(\boldsymbol{\lambda}, \boldsymbol{\phi} \mid \boldsymbol{x}) \, d\boldsymbol{\phi} \propto p(\boldsymbol{x} \mid \boldsymbol{\lambda}) \int_{\boldsymbol{\phi}} p(\boldsymbol{\lambda} \mid \boldsymbol{\phi}) \, p(\boldsymbol{\phi}) \, d\boldsymbol{\phi}$.

The EB solution avoids having to deal with the marginalization operation over $\boldsymbol{\phi}$ by estimating it from the data using maximum likelihood (or method of moments) applied to the marginal likelihood, $p(\boldsymbol{x} \mid \boldsymbol{\phi}) = \int_{\boldsymbol{\lambda}} p(\boldsymbol{x} \mid \boldsymbol{\lambda}) \, p(\boldsymbol{\lambda} \mid \boldsymbol{\phi}) \, d\boldsymbol{\lambda}$. The estimate $\hat{\boldsymbol{\phi}}$ is plugged into the prior distribution, $p(\boldsymbol{\lambda} \mid \hat{\boldsymbol{\phi}})$, which is then used to perform Bayesian inference in the usual way with the posterior, $p(\boldsymbol{\lambda} \mid \boldsymbol{x}) \propto p(\boldsymbol{x} \mid \boldsymbol{\lambda}) \, p(\boldsymbol{\lambda} \mid \hat{\boldsymbol{\phi}})$. So, essentially, the EB approach replaces an integration operation by a maximization operation applied to $p(\boldsymbol{x} \mid \boldsymbol{\phi})$.

### 2.2.1. APPROXIMATE MLE USING NPE-PFN

To implement EB under the NPE-PFN framework it is necessary to approximate the MLE estimate of $\phi$ using NPE-PFN machinery. The approach is described in full detail in Appendix E, but the basic idea is to perform over-confident large sample Bayesian inference using multiple copies of the observed data $\boldsymbol{x}_o$ and leverage the connection between MLE estimation and large sample Bayesian inference to approximate the MLE using NPE-PFN computations.

The overconfident asymptotic posterior distribution is obtained by replacing the observed data $\boldsymbol{x}_o$ by the much larger dataset $\boldsymbol{w}_o = \{\boldsymbol{x}_o, \ldots, \boldsymbol{x}_o\}$ during the computation of the posterior distribution (where $\boldsymbol{w}_o$ is constructed by concatenating multiple copies of $\boldsymbol{x}_o$). This posterior will be over-confident because despite the fact that $\boldsymbol{w}_o$ is much larger than $\boldsymbol{x}_o$, adding multiple copies of the same dataset (rather than drawing new samples from the true data generating process) do not provide any additional information about the parameters of interest. Hence, if $\boldsymbol{x}_o$ has $m$ samples and $\boldsymbol{w}_o$ is composed of $K$ copies of $\boldsymbol{x}_o$, the effective sample size of $\boldsymbol{w}_o$ is still $m$ even though it contains $n = mK$ samples. As a consequence, it strongly underestimates the true uncertainty of any posterior inferences.

It follows from standard Bayesian asymptotic theory (see Appendix E.2 for a review) that, as the sample size $n$ increases, the likelihood dominates the prior and the posterior distribution converges to a normal distribution centered at the MLE estimate, which converges to a degenerate point mass distribution around the true (or pseudo-true) parameter value as $n \to \infty$. But in the case of the overconfident posterior distribution the posterior is unable to converge to the true parameter value since the extra copies of the data do not provide any additional information. Rather, the overconfident posterior collapses around the MLE of the original data. These results are formalized in Theorems 4 and 5 in Appendix E.3.

### 2.2.2. NPE-PFN-BASED EB FOR TWO-LEVEL MODELS

In hierarchical models with grouped data each group is associated with its own parameter $\lambda_j$ yielding a likelihood of the form $p(\boldsymbol{x} \mid \boldsymbol{\lambda}) = \prod_j p(x_{ij} \mid \lambda_j)$, where the group-specific priors $p(\lambda_j \mid \boldsymbol{\phi})$ share a common hyperparameter, $\boldsymbol{\phi}$. Empirical Bayes (EB) methods often improve inference on the $\lambda_j$ by borrowing strength across groups through the estimation of $\boldsymbol{\phi}$. However, in flat likelihood settings where the estimation of $\boldsymbol{\phi}$ is difficult the EB approach will generally be less effective. One classical example is the estimation of the between group variance parameter in the hierarchical normal means model when the number of groups and the magnitude of the variance are small. (We will revisit this setting in the evaluations presented in section 4.)

To improve statistical inference in such flat likelihood set-tings rather than working with the "single-level" Bayesian hierarchical model $p(\boldsymbol{\lambda}, \boldsymbol{\phi} \mid \boldsymbol{x}_o) \propto p(\boldsymbol{x}_o \mid \boldsymbol{\lambda}) \, p(\boldsymbol{\lambda} \mid \boldsymbol{\phi}) \, p(\boldsymbol{\phi})$ we consider the two-level hierarchical Bayesian model, $p(\boldsymbol{\lambda}, \boldsymbol{\phi}_1, \boldsymbol{\phi}_2 \mid \boldsymbol{x}_o) \propto p(\boldsymbol{x}_o \mid \boldsymbol{\lambda}) p(\boldsymbol{\lambda} \mid \boldsymbol{\phi}_1) p(\boldsymbol{\phi}_1 \mid \boldsymbol{\phi}_2) p(\boldsymbol{\phi}_2)$ and apply the EB approach to the second-level prior, $p(\boldsymbol{\phi}_2)$. However, under our proposed NPE-PFN-based EB approach, instead of plugging-in an estimate $\hat{\boldsymbol{\phi}}_2$ into the first stage prior $p(\boldsymbol{\phi}_1 \mid \boldsymbol{\phi}_2)$ and working with the simplified model $p(\boldsymbol{\lambda}, \boldsymbol{\phi}_1 \mid \boldsymbol{x}_o) \propto p(\boldsymbol{x}_o \mid \boldsymbol{\lambda}) \, p(\boldsymbol{\lambda} \mid \boldsymbol{\phi}_1) \, p(\boldsymbol{\phi}_1 \mid \hat{\boldsymbol{\phi}}_2)$, we replace $p(\boldsymbol{\phi}_2)$ by an overconfident posterior $p(\boldsymbol{\phi}_2 \mid \boldsymbol{w}_o)$. Since this overconfident posterior becomes highly concentrated around the MLE estimate $\hat{\boldsymbol{\phi}}_2$ as the number of copies of $\boldsymbol{x}_o$ in $\boldsymbol{w}_o$ increase, in practice this prior replacement approach closely approximates the plugging-in operation.

Intuitively, when the data provides limited information about $\boldsymbol{\phi}_1$, it follows that replacing $p(\boldsymbol{\phi}_2)$ by the more informative distribution $p(\boldsymbol{\phi}_2 \mid \boldsymbol{w}_o)$ can potentially improve the inferences about $\boldsymbol{\phi}_1$ because $p(\boldsymbol{\phi}_2 \mid \boldsymbol{w}_o)$ contains some information about the data $\boldsymbol{x}_o$.

Algorithms 4 and 5 in Appendix E.3 describe the implementation of this strategy using the NPE-PFN framework. The basic idea is to first call Algorithm 4 to generate the overconfident posterior $p(\boldsymbol{\phi}_2 \mid \boldsymbol{w}_o)$ and then use the samples from this distribution as the start point for the generation of the training data used by the NPE-PFN approach.

Finally, it is important to highlight that while this "level-two" EB strategy focusing on $\boldsymbol{\phi}_2$ can be trivially implemented under the NPE-PFN paradigm, it is generally quite difficult to implement using alternative statistical frameworks.

## 3. Experiments: MAP/MLE equivalence

Here, we evaluate how well the theoretical results in Theorems 1 and 2 hold in practice for approximate posterior distributions generated by the NPE-PFN approach.

Our evaluations included 6 widely used univariate probability models, $p(x \mid \lambda)$, with scalar valued $\lambda$ parameter and closed form MLE estimators, $\hat{\lambda}_{ML}$. These included 4 models with unbounded parameter spaces (poisson, exponential, normal with known variance, and normal with known mean) and 2 models with bounded parameter spaces (bernoulli and negative binomial with fixed number of successes).

We evaluate the MAP/MLE equivalence performance with respect to sample size $(n)$ and number of training examples used by the NPE-PFN approach $(n_{tr})$. For each model we performed 9 separate simulation experiments encompassing all combinations of $n = \{30, 60, 120\}$ and $n_{tr} = \{250, 500, 1000\}$. Each simulation experiment was replicated 30 times with data generated with distinct $\lambda$ values. (See Appendix F.1 for further simulation details.)

Performance was measured by computing the relative ab-

solute distance (RAD) between the MLE estimate and the mode of the approximate posterior distribution generated with the NPE-PFN methodology, as described in equation 24 in Appendix F.2.

Table 1 report the results pulled across all 9 simulation settings. Across all models, the average relative absolute distance between the MLE and the posterior mode varied between 1.5% and 4.5% of the range of the posterior. This suggests that MAP inferences based on the approximate posteriors approximate well the MLE inferences in practice.

*Table 1.* Relative absolute distances across models. Abbreviations: poisson (PO), exponential (EX), normal with known variance (NM), normal with known mean (NV), bernoulli (BE), negative binomial (NB). Sd stands for standard deviation.

| MODEL | P0 | EX | NM | NV | BE | NB |
|---|---|---|---|---|---|---|
| MEAN | 0.025 | 0.045 | 0.017 | 0.035 | 0.015 | 0.031 |
| SD | 0.019 | 0.035 | 0.014 | 0.028 | 0.011 | 0.025 |

Appendix F.2 presents more detailed analyses of these results. Appendix G presents direct performance comparisons between uniform-half-t and reference priors.

## 4. Experiments: NPE-PFN-based EB

For this evaluation we consider the classical hierarchical normal means model (also known as the random intercepts model) given by,

$$p(x_{ij} \mid \mu, \alpha_j, \sigma) \sim N(\mu + \alpha_j \,,\, \sigma^2)\,,\, p(\alpha_j \mid \tau) \sim N(0\,,\, \tau^2)\,,$$

for $i = 1, \ldots, n_j$, $j = 1, \ldots, J$, where $J$ represents the number of groups and $\mu$, $\alpha_j$, $\sigma^2$, and $\tau^2$ represent, respectively, the population mean, the group level effects, the residual variance, and the between groups variance.

When $J$ and $\tau^2$ are small it can be very challenging to estimate $\tau$ (Gelman, 2006), and the current state-of-the-art baseline solution to this problem is to adopt a weakly informative prior for $\tau$. For our evaluations we adopt half-t distributions for the variance component parameters $\tau$ and $\sigma$, and a t-distribution for $\mu$. The corresponding full Bayesian hierarchical model is described in equations 31 to 35 in Appendix H.1. We denote this model by BASE, and implement it via MCMC using the `brms` R package.

Our proposed two-stage hierarchical model (based on hierarchical uniform-half-t priors) is described in equations 36 to 43. The EB model is obtained by replacing the second stage priors by the respective overconfident posterior distributions, as described in section 2.2.2 (see equations 44 and 45 in Appendix H.1).

For this experiment, we simulated data containing $J = 3$ groups, with small between group variance. (Note that we restrict our evaluations to this "small $J$ and small $\tau$" setting because this represents a very challenging setting for the

estimation of $\tau$ parameter. Settings with larger number of groups and/or larger between group variances are not problematic and are not investigated in our evaluations.)

Additional simulation set-up details are provided in Appendix H.2. The goal is to make inferences about the population parameters $\mu$, $\sigma$, and $\tau$. Figure 1 report the MSE results comparing the BASE (red) and EB-NPE-PFN (purple). Panels a and b show that (as expected) the performance of the 2 methods were very close for the $\mu$ and $\sigma$ parameters (as the likelihood is informative for these 2 parameters). Panel c shows that the BASE method has difficulty to estimate the $\tau$ parameter and produce relatively high MSE scores (note the larger dynamic range in the y-axis). The EB-NPE-PFN approach (purple boxplot), on the other hand, was able to generate considerably more accurate estimates.

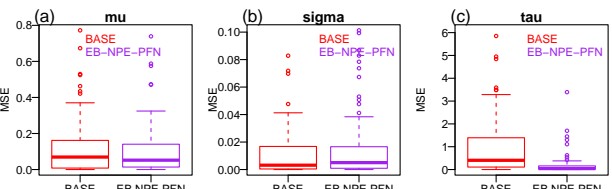

*Figure 1.* MSE between true and estimated parameter values.

To further highlight the contribution of the EB approach to the estimation of $\tau$, we present in Figure 4 (Appendix H.3) an ablation study where the EB-NPE-PFN approach is replaced by a NPE-PFN approach using the exact same two-stage hierarchial Bayesian model. Without the EB component the NPE-PFN performance is comparable to the BASE method.

Finally, observe that the EB-NPE-PFN results reported here were generated using the parameter order in equation 46. Different parameter orderings generated similar results.

## 5. Conclusions

This paper describes a new application of TFMs. Objective and data-driven Bayesian methods play a central role in scientific inference by providing principled, default analyses that enable coherent uncertainty quantification when substantive prior information is limited.

This paper primarily focuses on the methodological contributions to objective Bayesian inference and data-driven Bayesian analysis described in Section 2. The experiments in Section 3 are meant to demonstrate that the approximate posteriors generated by NPE-PFN closely recover the theoretical results of Theorems 1 and 2. The experiments in Section 4 further illustrate the flexibility of NPE-PFN by addressing a problem that is considerably more challenging under traditional Bayesian computation frameworks.

Limitations of the present work are listed in Appendix I.

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

# Appendix

# Contents

# A. Related work and contribution positioning

PFN models (Muller et al., 2022) have been applied to multiple domains including supervised learning (Hollmann et al., 2023; Muller et al., 2025; Hollmann et al., 2025), time series forecasting (Verdenius et al., 2024; Hoo et al., 2025; Bhethanabhotla et al., 2024), Bayesian optimization (Muller et al., 2023), and synthetic data generation (Ma et al. 2023; Chaibub Neto, 2026).

## A.1. Bayesian statistical inference with tabular PFNs

Recent work has also explored tabular PFNs for performing Bayesian statistical inference. Vetter et al. (2025) proposed NPE-PFN, which repurposed the pre-trained TabPFN model as an autoregressive generator for simulation-based inference, enabling training-free posterior estimation and sampling. In parallel, Reuter et al. (2025) studied whether transformers can learn full Bayesian inference by training transformer models with flow matching to generate posterior samples from classical latent-variable statistical models such as generalized linear models, factor analysis, and Gaussian mixture models. Together, these works suggest that foundation models and in-context learning can serve as reusable inference engines, reducing reliance on per-task MCMC, variational approximations, or simulator-specific neural posterior estimators.

These approaches differ, however, in modeling strategy and target problem. NPE-PFN adapts an already pre-trained TabPFN model for posterior inference, making the method essentially training-free for the user, and focuses on simulation-based inference, especially scientific inverse problems where the likelihood is unavailable but simulation is possible. Reuter et al., by contrast, focus on Bayesian inference for more classical families of statistical models, training transformer-based inference models for each considered model class. This requires substantially more up-front training effort while still enabling in-context posterior sampling at deployment time.

In this paper, we describe how to perform objective Bayesian inference and empirical Bayes analysis using pre-trained tabular foundation models. Similarly to Reuter et al., our focus is on standard statistical models. However, following Vetter et al., we adopt the NPE-PFN framework as our modeling strategy. Application of the NPE-PFN framework to these problems requires, nonetheless, the development of new methodology.

## A.2. Objective Bayesian inference

Objective Bayesian inference aim at developing default priors that let the likelihood dominate the analysis. The most widely used objective Bayes approach is reference analysis (Bernardo, 1979; Bernardo and Smith, 1994). This approach adopts an information theoretic criterion where the objective priors are constructed by maximizing the expected Kullback-Leibler divergence between the prior and the posterior, thereby selecting the prior that allows the data to have the greatest possible influence on inference. Direct implementation of reference analysis using the NPE-PFN approach can, nonetheless, be problematic. First, reference priors are often improper making it difficult to perform NPE-PFN calculations which require the ability to sample directly from the prior distributions. Second, reference priors are model specific, meaning that each distinct probability model has its own distinct reference prior. Third, the analytical derivation of reference priors can be difficult in practice, being sometimes intractable for some more complicated models.

Our newly proposed MAP/MLE equivalence criterion addresses all these difficulties. **First**, our uniform-half-t and uniform default priors are proper probability distributions, from which it is very easy to draw samples.

**Second**, our MAP/MLE equivalence default priors can be applied across arbitrary probability models. As described in detail in section 2.1, our default priors correspond, essentially, to three distinct prior types: one for probability models parameterized by parameters assuming values in the positive real line, $\mathbb{R}^+$; another for probability models with parameters assuming values in the entire real line, $\mathbb{R}$; and finally another for probability models with bounded parameter spaces assuming values in an arbitrary interval $[a_j, b_j]$. More specifically, for any arbitrary probability model $p(\boldsymbol{x} \mid \boldsymbol{\lambda})$ parameterized by $\boldsymbol{\lambda} = \{\lambda_1, \ldots, \lambda_l\}$, we have that for each separate $\lambda_j$ parameter the MAP/MLE equivalence prior corresponds to:

- A hierarchical uniform-half-t prior, $p(\lambda_j \mid \phi_j)p(\phi_j) = U(0, \phi_j)\, ht_{d_{\phi_j}}(s^2_{\phi_j})$, when $\lambda_j \in \mathbb{R}^+$. Note that this exact same prior can be used to achieve MAP/MLE equivalence for multiple models including, for instance, the poisson, exponential, gamma, inverse-gamma, beta, and normal (with known mean) models.

- A hierarchical uniform-half-t prior $p(\lambda_j \mid \phi_j)p(\phi_j) = U(-\phi_j, \phi_j)\, ht_{d_{\phi_j}}(s^2_{\phi_j})$, when $\lambda_j \in \mathbb{R}$. Note that this exact same prior can be used to achieve MAP/MLE equivalence in (for instance) location-scale models with known scale

parameters. These include, for example, the normal, student-t, laplace, and logistic models (when the scale prameter is known).

- A uniform prior $p(\lambda_j) = U(a_j, b_j)$, when $\lambda_j \in [a_j, b_j]$. Note that this exact same prior provides MAP/MLE equivalence for probability models parameterized by probability of success parameters such as the bernoulli, binomial, and negative binomial models.

Clearly, for multi-parameter probability models we can mix and match these three distinct prior components. For example, for a normal model with unknown mean and variance, we have that $\boldsymbol{\lambda} = \{\lambda_1, \lambda_2\} = \{\mu, \sigma^2\}$ and $p(\boldsymbol{x} \mid \boldsymbol{\lambda}) = p(\boldsymbol{x} \mid \mu, \sigma^2)$ and the MAP/MLE equivalence prior is given by,

$$p(\boldsymbol{\lambda} \mid \boldsymbol{\phi})\, p(\boldsymbol{\phi}) = \prod_{j=1}^{2} p(\lambda_j \mid \phi_j)\, p(\phi_j) = U(-\phi_1, \phi_1)\, ht_{d_{\phi_1}}(s_{\phi_1}^2) \times U(0, \phi_2)\, ht_{d_{\phi_2}}(s_{\phi_2}^2)\,, \tag{2}$$

where $ht_{d_{\phi_j}}(s_{\phi_j}^2)$ represents a half-t distribution with scale parameter $s_{\phi_j}^2$.

**Third**, contrary to reference priors which might not be available for more intractable probability models, objective Bayes analysis based on the MAP/MLE equivalence criterion can be applied to any probability model from which we can draw samples. Since this represents an easier requirement, in practice, our objective Bayes criterion is more generally applicable than reference analysis.

Finally, observe that while reference analysis aims to generate priors that maximize the expected Kullback-Leibler divergence between the prior and the posterior, in practice, this criterion is closely related to our proposed MAP/MLE equivalence criterium since maximizing the expected KL-divergence between prior and posterior effectively selects a prior that let the likelihood dominate the analysis. Hence, in practice we see that the expectation of reference posterior distributions are often closely related (or exactly match) the MLE estimate from the associated probability model.

### A.3. Empirical Bayes

Empirical Bayes (EB) analysis provide a principled compromise between fully Bayesian and frequentist approaches. By employing a data-driven prior, this procedure can also be considered objective in the sense that it reduces reliance on subjective elicitation by learning an informative prior directly from data.

Direct implementation of EB analyses under the NPE-PFN framework is (at first sight) problematic because NPE-PFN computations are Bayesian in nature, whereas EB approaches require MLE estimation of parameters from the marginal probability model (which corresponds to a frequentist calculation outside the purview of NPE-PFN Bayesian computations). Our proposed solution, described in section 2.2, leverages the connection between MLE and asymptotic Bayesian theory to approximate MLE estimation using overconfident asymptotic Bayesian posteriors generated with NPE-PFN machinery.

## B. Background

Vetter et al. (2025) described how to perform Bayesian inference using tabular foundation models in the context of simulation based inference, focusing on the TabPFN model (Hollmann et al., 2025).

TabPFN is a state-of-the-art foundation model for tabular data, pre-trained on an extensive corpus of synthetic datasets that encompass diverse feature types, noise structures, and functional dependencies. By internalizing these variations, the model leverages a robust and transferable prior over tabular distributions. Formally grounded as a Prior-data Fitted Network (PFN) (Muller et al., 2022), TabPFN performs amortized Bayesian prediction by leveraging the prior induced during its synthetic pre-training phase. This architecture effectively treats the neural network as an inference engine that approximates Bayesian prediction through in-context learning (ICL) (Brown et al., 2020). At inference time, the pre-trained foundation model employs training features, $\boldsymbol{z}_{tr}$, and training targets, $y_{tr}$, as the "context" data, and uses the test set features, $\boldsymbol{z}_{ts}$, to "query" the model for the target predictions. The output of the query is a sample/prediction $\hat{y}_{ts}$ from the posterior predictive distribution of $y_{ts}$, $p(y_{ts} \mid \boldsymbol{z}_{ts}, \boldsymbol{z}_{tr}, y_{tr})$, generated by a single forward pass through the model's neural network. Throughout this paper we represent the transformer-based TabPFN model used to approximate $p(y_{ts} \mid \boldsymbol{z}_{ts}, \boldsymbol{z}_{tr}, y_{tr})$ by $q_{\boldsymbol{\psi}}(y_{ts} \mid \{\boldsymbol{z}_{ts}\}, \{y_{tr}, \boldsymbol{z}_{tr}\})$, where $\boldsymbol{\psi}$ represents the neural network model parameters learned during pre-training.

Vetter et al. (2025) describes how to repurpose TabPFN's probabilistic inferential engine for performing amortized Bayesian inference, in an approach denoted as Neural Posterior Estimation with Prior-data Fitted Networks (NPE-PFN). In the

simplest case where $\boldsymbol{\theta}$ is an scalar parameter, the basic idea is to treat $\theta$ as the target variable and treat $\boldsymbol{x}$ as the feature data in the TabPFN model. To this end, the training data (used as the context examples in the TabPFN model) is generated by: (i) sampling $\theta$ values from its prior distribution $p(\theta)$, $\theta_{tr} \sim p(\theta)$; and (ii) sampling $\boldsymbol{x}$ values (conditional on $\theta_{tr}$ values) from the adopted probability model, $p(\boldsymbol{x}_{tr} \mid \theta_{tr})$. Then, samples from the posterior distribution $p(\theta \mid \boldsymbol{x}_o)$ are generated by using the observed data $\boldsymbol{x}_o$ to query the TabPFN model for a prediction of $\theta$,

$$\theta_{ts} \sim q_{\boldsymbol{\psi}}(\theta \mid \{\boldsymbol{x}_o\}, \{\theta_{tr}, \boldsymbol{x}_{tr}\}) . \tag{3}$$

In the general case where $\boldsymbol{\theta}$ is vector valued, Vetter et al. (2025) proposed to use TabPFN autoregressively, that is, to use it to sequentially predict the next dimension of $\boldsymbol{\theta}$ conditional on the previously processed dimensions. For instance, in the case where $\boldsymbol{\theta} = \{\lambda, \phi\}$ the posterior distribution is approximated by,

$$p(\lambda, \phi \mid \boldsymbol{x}_o) \approx q_{\boldsymbol{\psi}}(\phi \mid \{\boldsymbol{x}_o\}, \{\phi_{tr}, \boldsymbol{x}_{tr}\}) \, q_{\boldsymbol{\psi}}(\lambda \mid \{\phi_{ts}, \boldsymbol{x}_o\}, \{\phi_{tr}, \lambda_{tr}, \boldsymbol{x}_{tr}\}) , \tag{4}$$

where $\phi$ is processed first and $\lambda$ is processed conditional on $\phi$. (Or, equivalently, by processing $\lambda$ first, and then $\phi$ conditional on $\lambda$.)

In general, for $\boldsymbol{\theta} = \{\theta_1, \ldots, \theta_l\}$ with dimension $l$ the posterior is approximated by,

$$p(\theta_1, \ldots, \theta_l \mid x_o) \approx \prod_{j=1}^{l} q_{\boldsymbol{\psi}}(\theta_j \mid \{\theta_1^{ts}, \ldots, \theta_{j-1}^{ts}, \boldsymbol{x}_o\}, \{\theta_1^{tr}, \ldots, \theta_j^{tr}, \boldsymbol{x}_{tr}\}) . \tag{5}$$

Algorithm 1 describes the implementation of the NPE-PFN method for a single-stage hierarchical prior parameterized as $p(\lambda \mid \phi) \, p(\phi)$ (such as an uniform-half-t prior described in section 2.1). For the sake of notational simplicity, the algorithm describes the implementation for scalar valued $\lambda$ and $\phi$ parameters. In the case where $\lambda$ and $\phi$ are multidimensional, we replace the posterior generation steps (which in Algorithm 1 correspond to lines 5 and 6) by for-loops iterating over each dimension of the parameter vector according to a factorization analogous to the one described in equation 5.

---

**Algorithm 1** NPE-PFN for single-level hierarchical priors

---

1: **Input:** data vector, $x_o$; probability model, $p(x \mid \lambda)$; prior distribution, $p(\lambda \mid \phi) \, p(\phi)$; number of training samples, $n_{tr}$; number of test (posterior) samples, $n_{ts}$

2: $\phi_{tr} \sim^{n_{tr}} p(\phi)$ {Draw $n_{tr}$ samples of $\phi$ from the hyper-prior $p(\phi)$.}

3: $\lambda_{tr} \sim^{n_{tr}} p(\lambda \mid \phi_{tr})$ {Draw $n_{tr}$ samples of $\lambda$ conditional on the values of $\phi_{tr}$ generated in the previous step.}

4: $x_{tr} \sim^{n_{tr}} p(x \mid \lambda_{tr})$ {Draw $n_{tr}$ samples of $x$ conditional on the values of $\lambda_{tr}$ generated in the previous step.}

5: $\lambda_{ts} \sim^{n_{ts}} q_{\psi}(\lambda \mid \{x_o\}, \{\lambda_{tr}, x_{tr}\})$ {Draw $n_{ts}$ samples from the posterior distribution of $\lambda$ using a PFN model trained on $\{\lambda_{tr}, x_{tr}\}$ and queried on $x_o$.}

6: $\phi_{ts} \sim^{n_{ts}} q_{\psi}(\phi \mid \{\lambda_{ts}, x_o\}, \{\phi_{tr}, \lambda_{tr}, x_{tr}\})$ {Draw $n_{ts}$ samples from the posterior distribution of $\phi$ using a PFN model trained on $\{\phi_{tr}, \lambda_{tr}, x_{tr}\}$ and queried on $\{\lambda_{ts}, x_o\}$.}

7: **Output:** $n_{ts}$ samples $\lambda_{ts}$ and $\phi_{ts}$ from the posterior distribution of $\lambda$ and $\phi$.

---

Similarly, Algorithm 2 presents the NPE-PFN algorithm for a two-stage hierarchical prior with format, $p(\lambda \mid \phi_1) \, p(\phi_1 \mid \phi_2) \, p(\phi_2)$ (such as the hierarchical prior used in the EB-NPE-PFN approach described in section 2.2.2).

The next section describes an important implementation detail.

## C. Implementation details

The TabPFN model does not automatically set any constraints in the range of values it can generate. Hence, direct application of TabPFN for probabilistic inference may generate values outside the support of a given parameter variable. For instance, TabPFN can potentially generate negative values when drawing samples from the posterior distribution of a scale parameter.

To circumvent this problem, we first transform the input parameter data to values in the real-line before it is fed to the TabPFN model, and then transform the output of the TabPFN model back to the original parameter support.

For example, for any $\lambda_j$ parameter such that $\lambda_j \in \mathbb{R}^+$, we first transform $\lambda_j$ to $\log \lambda_j$ before feeding the data to TabPFN (so that the transformed $\log \lambda_j$ variable can assume values in the entire real line, $\mathbb{R}$), and then transform the output from TabPFN back to the $\mathbb{R}^+$ range using the inverse $\exp(\log \lambda_j)$.

---

**Algorithm 2** NPE-PFN for two-level hierarchical priors

---

1: **Input:** data vector, $x_o$; probability model, $p(x \mid \lambda)$; prior distribution, $p(\lambda \mid \phi_1) \, p(\phi_1 \mid \phi_2) \, p(\phi_2)$; number of training samples, $n_{tr}$; number of test (posterior) samples, $n_{ts}$

2: $\phi_{2,tr} \sim^{n_{tr}} p(\phi_2)$ {Draw $n_{tr}$ samples of $\phi_2$ from the hyper-prior $p(\phi_2)$.}

3: $\phi_{1,tr} \sim^{n_{tr}} p(\phi_1 \mid \phi_{2,tr})$ {Draw $n_{tr}$ samples of $\phi_1$ conditional on the values of $\phi_{2,tr}$ generated in the previous step.}

4: $\lambda_{tr} \sim^{n_{tr}} p(\lambda \mid \phi_{1,tr})$ {Draw $n_{tr}$ samples of $\lambda$ conditional on the values of $\phi_{1,tr}$ generated in the previous step.}

5: $x_{tr} \sim^{n_{tr}} p(x \mid \lambda_{tr})$ {Draw $n_{tr}$ samples of $x$ conditional on the values of $\lambda_{tr}$ generated in the previous step.}

6: $\lambda_{ts} \sim^{n_{ts}} q_\psi\big(\lambda \mid \{x_o\}, \{\lambda_{tr}, x_{tr}\}\big)$ {Draw $n_{ts}$ samples from the posterior distribution of $\lambda$ using a PFN model trained on $\{\lambda_{tr}, x_{tr}\}$ and queried on $x_o$.}

7: $\phi_{1,ts} \sim^{n_{ts}} q_\psi\big(\phi_1 \mid \{\lambda_{ts}, x_o\}, \{\phi_{1,tr}, \lambda_{tr}, x_{tr}\}\big)$ {Draw $n_{ts}$ samples from the posterior distribution of $\phi_1$ using a PFN model trained on $\{\phi_{1,tr}, \lambda_{tr}, x_{tr}\}$ and queried on $\{\lambda_{ts}, x_o\}$.}

8: $\phi_{2,ts} \sim^{n_{ts}} q_\psi\big(\phi_2 \mid \{\phi_{1,ts}, \lambda_{ts}, x_o\}, \{\phi_{2,tr}, \phi_{1,tr}, \lambda_{tr}, x_{tr}\}\big)$ {Draw $n_{ts}$ samples from the posterior distribution of $\phi_2$ using a PFN model trained on $\{\phi_{2,tr}, \phi_{1,tr}, \lambda_{tr}, x_{tr}\}$ and queried on $\{\phi_{1,ts}, \lambda_{ts}, x_o\}$.}

9: **Output:** $n_{ts}$ samples $\lambda_{ts}$, $\phi_{1,ts}$, and $\phi_{2,ts}$ from the posterior distribution of $\lambda$, $\phi_1$ and $\phi_2$.

---

Similarly, for any $\lambda_j \in [0, 1]$, we adopt the logit transformation,

$$\mathtt{logit}(x) = \log\left(\frac{x}{1-x}\right) , \tag{6}$$

to transform its support to the $\mathbb{R}$ range, and the inverse logit transformation, $\mathtt{invlogit}()$,

$$\mathtt{invlogit}(x) = \frac{1}{1 - \exp(-x)} \tag{7}$$

to transform it back to the $[0, 1]$ range.

# D. Proofs of Theorems 1 and 2

## D.1. Proof of Theorem 1

*Proof.* First, note that if we adopt a uniform prior for $p(\lambda_j \mid \phi_j)$, then the hyperprior $p(\phi_j)$ must have support over the entire positive real line ($\phi_j > 0$) to ensure that the hierarchical prior does not restrict the support of $\lambda_j$ in the posterior distribution. To see this, consider the case where $\lambda_j > 0$. Since $\lambda_j$ is finite, there always exists some $\phi_j$ such that $\lambda_j < \phi_j$. Therefore, if we specify $p(\lambda_j \mid \phi_j)$ as a $U(0, \phi_j)$ prior, the hyperprior $p(\phi_j)$ must assign positive probability to all $\phi_j > 0$ so that $\lambda_j$ can take any value on the positive real line.

An analogous argument applies when $\lambda_j \in \mathbb{R}$. In this setting, using a $U(-\phi_j, \phi_j)$ prior for $p(\lambda_j \mid \phi_j)$ again requires $p(\phi_j)$ to have support on $\phi_j > 0$, ensuring that $\lambda_j$ can assume any value on the real line.

Now, let $p(\phi_j)$ be an arbitrary prior with support $\phi_j > 0$ and $p(\lambda_j \mid \phi_j) \sim U(a, \phi_j)$, where $a = 0$ if $\lambda_j > 0$, and $a = -\phi_j$ if $-\infty < \lambda_j < \infty$. Under this hierarchical prior, the marginal posterior distribution of $\boldsymbol{\lambda}$ is given by,

$$
\begin{aligned}
p(\boldsymbol{\lambda} \mid \boldsymbol{x}) &= \int_{\boldsymbol{\phi}} p(\boldsymbol{\lambda}, \boldsymbol{\phi} \mid \boldsymbol{x}) \, d\boldsymbol{\phi} = \frac{p(\boldsymbol{x} \mid \boldsymbol{\lambda})}{p(\boldsymbol{x})} \int_{\boldsymbol{\phi}} p(\boldsymbol{\lambda} \mid \boldsymbol{\phi}) \, p(\boldsymbol{\phi}) \, d\boldsymbol{\phi} \\
&= \frac{p(\boldsymbol{x} \mid \boldsymbol{\lambda})}{p(\boldsymbol{x})} \prod_{j=1}^{l} \int_{\phi_j} p(\lambda_j \mid \phi_j) \, p(\phi_j) \, d\phi_j = p(\boldsymbol{x} \mid \boldsymbol{\lambda}) \underbrace{\frac{1}{p(\boldsymbol{x})} \prod_{j=1}^{l} \int_{\phi_j} \frac{1}{\phi_j - a} \, p(\phi_j) \, d\phi_j}_{c} \\
&= c \, p(\boldsymbol{x} \mid \boldsymbol{\lambda})
\end{aligned}
\tag{8}
$$

where the normalization constant $c$ does not depend on $\boldsymbol{\lambda}$.

Then the maximum a posteriori (MAP) of the posterior distribution $p(\boldsymbol{\lambda} \mid \boldsymbol{x})$,

$$\underset{\boldsymbol{\lambda}}{\operatorname{argmax}} \, p(\boldsymbol{\lambda} \mid \boldsymbol{x}) = \underset{\boldsymbol{\lambda}}{\operatorname{argmax}}[c \, p(\boldsymbol{x} \mid \boldsymbol{\lambda})] = \underset{\boldsymbol{\lambda}}{\operatorname{argmax}} \, p(\boldsymbol{x} \mid \boldsymbol{\lambda}) \tag{9}$$

equals the maximum likelihood estimator, $\hat{\boldsymbol{\lambda}}_n = \mathrm{argmax}_{\boldsymbol{\lambda}}\, p(\boldsymbol{x} \mid \boldsymbol{\lambda})$, since multiplying $p(\boldsymbol{x} \mid \boldsymbol{\lambda})$ by the constant $c$ does not change the maximizer of $p(\boldsymbol{x} \mid \boldsymbol{\lambda})$. $\qquad\square$

### D.2. Proof of Theorem 2

*Proof.* Under the uniform prior, $p(\boldsymbol{\lambda}) = \prod_{j=1}^l p(\lambda_j)$, where $p(\lambda_j) \sim U(a_j, b_j)$, the posterior distribution of $\boldsymbol{\lambda}$ is given by,

$$p(\boldsymbol{\lambda} \mid \boldsymbol{x}) = \frac{p(\boldsymbol{x} \mid \boldsymbol{\lambda})}{p(\boldsymbol{x})} p(\boldsymbol{\lambda}) = p(\boldsymbol{x} \mid \boldsymbol{\lambda}) \underbrace{\frac{1}{p(\boldsymbol{x})} \prod_{j=1}^l \frac{1}{b_j - a_j}}_{c} = c\, p(\boldsymbol{x} \mid \boldsymbol{\lambda}) \tag{10}$$

where the normalization constant $c$ does not depend on $\boldsymbol{\lambda}$. Hence, maximizing the posterior distribution is equivalent to maximizing the likelihood, since the constant $c$ does not change the maximizer of the likelihood. $\qquad\square$

## E. Approximate maximum likelihood estimation using NPE-PFN - extended description

Here, we describe in detail how to perform MLE estimation using NPE-PFN machinery. As described in section 2.2.1 in the main text, the key idea is to generate an overconfident posterior distribution by working with the much larger dataset $\boldsymbol{w}_o = \{\boldsymbol{x}_o, \ldots, \boldsymbol{x}_o\}$ (instead of $\boldsymbol{x}_o$) during the computation of the posterior distribution. However, because there are limits to the size of the datasets that can be fed into the TabPFN model, in practice, instead of working with the much larger $\boldsymbol{w}_o$ dataset directly, we take advantage of the sequential updating property of Bayesian inference and generate the overconfident posterior iteratively using sequential updates.

In the next section we describe in detail the sequential updating property of Bayesian inference. In section E.2 we provide background on Bayesian asymptotic theory. Finally, in section E.3 we describe how to perform MLE estimation using NPE-PFN machinery.

### E.1. The sequential updating property of Bayesian inference

A well known result in Bayesian statistics is the sequential updating property of Bayesian inference where the posterior from one data batch becomes the prior for the next, leading to the same final posterior as processing all data at once. Let $\boldsymbol{x}^1$, $\boldsymbol{x}^2, \ldots \boldsymbol{x}^K$ represent $K$ subsets of the full dataset $\boldsymbol{x} = \{\boldsymbol{x}^1, \boldsymbol{x}^2, \ldots, \boldsymbol{x}^k\}$.

Consider the notation,

$$p^k(\boldsymbol{\theta} \mid \boldsymbol{x}^k) = p(\boldsymbol{\theta} \mid \boldsymbol{x}^k, \boldsymbol{x}^{k-1}, \ldots, \boldsymbol{x}^1), \qquad p^k(\boldsymbol{x}^k) = p(\boldsymbol{x}^k \mid \boldsymbol{x}^{k-1}, \ldots, \boldsymbol{x}^1), \tag{11}$$

In the sequential processing of the data, we generate a posterior distribution of $\boldsymbol{\theta}$ by updating our prior knowledge about $\boldsymbol{\theta}$, expressed by the prior distribution $p(\boldsymbol{\theta})$, with the first data batch,

$$p^1(\boldsymbol{\theta} \mid \boldsymbol{x}^1) = \frac{p(\boldsymbol{x}^1 \mid \boldsymbol{\theta})}{p(\boldsymbol{x}^1)} p(\boldsymbol{\theta}) \tag{12}$$

followed by posterior distribution updates where the posterior distribution of data batch $k$ is computed using the posterior distribution of the previous batch as the prior,

$$p^k(\boldsymbol{\theta} \mid \boldsymbol{x}^k) = \frac{p(\boldsymbol{x}^k \mid \boldsymbol{\theta})}{p^k(\boldsymbol{x}^k)} p^{k-1}(\boldsymbol{\theta} \mid \boldsymbol{x}^{k-1}), \qquad k = 2, \ldots, K. \tag{13}$$

The sequential updating property of Bayesian inference establishes that,

$$p^K(\boldsymbol{\theta} \mid \boldsymbol{x}^K) = p(\boldsymbol{\theta} \mid \boldsymbol{x}^1, \boldsymbol{x}^2, \ldots, \boldsymbol{x}^K). \tag{14}$$

To derive equation 14 we will show that the sequential posterior distribution $p^K(\boldsymbol{\theta} \mid \boldsymbol{x}^K)$ and the standard posterior distribution $p(\boldsymbol{\theta} \mid \boldsymbol{x}^1, \boldsymbol{x}^2, \ldots, \boldsymbol{x}^K)$ share the same underlying expression. The result only requires the assumption that the datasets $\boldsymbol{x}^1, \boldsymbol{x}^2, \ldots, \boldsymbol{x}^K$ are independent of each other conditional on $\boldsymbol{\theta}$, and follow directly from Bayes' theorem.

Starting with $p^K(\boldsymbol{\theta} \mid \boldsymbol{x}^K)$, note that by the application of recursive substitutions of equation 13 on itself, allows us to re-express the sequential posterior distribution at iteration $K$ as,

$$p^K(\boldsymbol{\theta} \mid \boldsymbol{x}^K) = \frac{p(\boldsymbol{x}^K \mid \boldsymbol{\theta})}{p^K(\boldsymbol{x}^K)} \, p^{K-1}(\boldsymbol{\theta} \mid \boldsymbol{x}^{K-1}) \tag{15}$$

$$= \frac{p(\boldsymbol{x}^K \mid \boldsymbol{\theta})}{p^K(\boldsymbol{x}^K)} \frac{p(\boldsymbol{x}^{K-1} \mid \boldsymbol{\theta})}{p^{K-1}(\boldsymbol{x}^{K-1})} \, p^{K-2}(\boldsymbol{\theta} \mid \boldsymbol{x}^{K-2}) \tag{16}$$

$$= \frac{p(\boldsymbol{x}^K \mid \boldsymbol{\theta})}{p^K(\boldsymbol{x}^K)} \frac{p(\boldsymbol{x}^{K-1} \mid \boldsymbol{\theta})}{p^{K-1}(\boldsymbol{x}^{K-1})} \times \ldots \times \frac{p(\boldsymbol{x}^1 \mid \boldsymbol{\theta})}{p^1(\boldsymbol{x}^1)} \, p(\boldsymbol{\theta}) \tag{17}$$

$$= \frac{\prod_{k=1}^{K} p(\boldsymbol{x}^k \mid \boldsymbol{\theta})}{p(\boldsymbol{x}^K \mid \boldsymbol{x}^{K-1}, \ldots, \boldsymbol{x}^1) \, p(\boldsymbol{x}^{K-1} \mid \boldsymbol{x}^{K-2}, \ldots, \boldsymbol{x}^1) \times \ldots \times p(\boldsymbol{x}^1)} \, p(\boldsymbol{\theta}) \tag{18}$$

$$= \frac{\prod_{k=1}^{K} p(\boldsymbol{x}^k \mid \boldsymbol{\theta})}{p(\boldsymbol{x}^1, \boldsymbol{x}^2, \ldots, \boldsymbol{x}^K)} \, p(\boldsymbol{\theta}) \,, \tag{19}$$

where $p^K(\boldsymbol{x}^K)$ represents a simplified notation for $p^K(\boldsymbol{x}^K) = p(\boldsymbol{x}^K \mid \boldsymbol{x}^{K-1}, \ldots, \boldsymbol{x}^1)$.

Now, moving to the standard posterior distribution, note that it can also be re-expressed as,

$$p(\boldsymbol{\theta} \mid \boldsymbol{x}^1, \boldsymbol{x}^2, \ldots, \boldsymbol{x}^K) = \frac{p(\boldsymbol{x}^1, \boldsymbol{x}^2, \ldots, \boldsymbol{x}^K \mid \boldsymbol{\theta})}{p(\boldsymbol{x}^1, \boldsymbol{x}^2, \ldots, \boldsymbol{x}^K)} \, p(\boldsymbol{\theta}) \tag{20}$$

$$= \frac{\prod_{k=1}^{K} p(\boldsymbol{x}^k \mid \boldsymbol{\theta})}{p(\boldsymbol{x}^1, \boldsymbol{x}^2, \ldots, \boldsymbol{x}^K)} \, p(\boldsymbol{\theta}) \,, \tag{21}$$

where the second equality follows from the assumption that the datasets $\boldsymbol{x}^1, \boldsymbol{x}^2, \ldots, \boldsymbol{x}^K$ are independent of each other conditional on $\boldsymbol{\theta}$.

### E.2. Asymptotic theory for the posterior distribution

The asymptotic behavior of the posterior distribution is well characterized in both the correctly specified and misspecified cases. The probability model $p(\boldsymbol{x} \mid \boldsymbol{\theta})$ is correctly specified when there exists a $\boldsymbol{\theta} = \boldsymbol{\theta}_*$ for which the true data distribution of the data, $f(\boldsymbol{x})$, is equal to $p(\boldsymbol{x} \mid \boldsymbol{\theta}_*)$. In contrast, the probability model is misspecified if $f(\boldsymbol{x}) \neq p(\boldsymbol{x} \mid \boldsymbol{\theta})$ for all $\boldsymbol{\theta} \in \boldsymbol{\Theta}$. See Gelman et al. (2010) for an introductory review and Bernardo and Smith (1994) and Schervish (1995) for more detailed accounts of large sample Bayesian inference. Result 3 summarizes the key theoretical result needed in this paper.

**Result 3. *Asymptotic normality of the posterior.* Let $\boldsymbol{x} = \{x_1, \ldots, x_n\}$. As $n$ increases, the posterior distribution, $p(\boldsymbol{\theta} \mid \boldsymbol{x})$, converges (under regularity conditions) to a normal distribution centered at the MLE estimator $\hat{\boldsymbol{\theta}}_n$ and with (co)variance inversely proportional to $n$.**

When the model is correctly specified, the (co)variance of the asymptotic posterior is given by $n^{-1} I^{-1}(\boldsymbol{\theta}_*)$, where $I(\boldsymbol{\theta}_*)$ corresponds to the model's Fisher information. When it is misspecified, the covariance corresponds to $n^{-1} A$, where $A$ is derived in Kleijn and van der Vaart (2012). The exact expression for the asymptotic (co)variances is unimportant for our purposes in this paper. The only relevant point is to note that it shrink towards 0 and the posterior gets increasingly more concentrated around $\hat{\boldsymbol{\theta}}_n$ as $n \to \infty$.

The regularity conditions are detailed in Section 7.4.2 of Schervish (1995) for the correctly specified case and in Kleijn and van der Vaart (2012) for the misspecified one. At a high level they ensure that the probability model (likelihood) is smooth and well behaved, that the empirical log-likelihood converges uniformly to a deterministic limit with a unique, well-separated maximizer (the true parameter under correct specification, or the KullbackLeibler minimizer under misspecification), and that the prior assigns positive mass in a neighborhood of this maximizer.

### E.3. Approximating MLE estimation using NPE-PFN

Here we describe how to approximate maximum likelihood estimation of a probability model using the NPE-PFN approach. (This technique will be needed in section 2.2.2, where we describe how to approximate empirical Bayes solutions using NPE-PFN machinery.) The basic idea is to perform over-confident large sample Bayesian inference using multiple copies

of the observed data $\boldsymbol{x}_o$ and leverage the connection between maximum likelihood estimation (MLE) and large sample Bayesian inference to approximate the MLE using NPE-PFN models.

Algorithm 3 describes the process in general terms while Algorithm 4 describes the corresponding implementation using the NPE-PFN approach for the two-level model. Note that these algorithms implement large sample inference using the sequential updating property of Bayesian inference described in detail in Appendix E.1. In our implementation in Algorithm 4 we adopt a sequential strategy rather than relying on a single processing step of a larger dataset because the underlying TabPFN model has a soft limit on the dataset sizes it can handle.

---

**Algorithm 3** Overconfident posterior distribution

1: **Input:** data vector, $\boldsymbol{x}_o$; number of iterations, $K$; probability model, $p(\boldsymbol{x} \mid \boldsymbol{\theta})$; prior distribution, $p(\boldsymbol{\theta})$
2: $\boldsymbol{x}_o^1 \leftarrow \boldsymbol{x}_o$ {Create a copy of $\boldsymbol{x}_o$.}
3: $p^1(\boldsymbol{\theta} \mid \boldsymbol{x}_o^1) = \dfrac{p(\boldsymbol{x}_o^1 \mid \boldsymbol{\theta})}{p^1(\boldsymbol{x}_o^1)} p(\boldsymbol{\theta})$
4: **for** $k \in 2 : K$ **do**
5: $\quad \boldsymbol{x}_o^k \leftarrow \boldsymbol{x}_o$ {Create a copy of $\boldsymbol{x}_o$.}
6: $\quad p^k(\boldsymbol{\theta} \mid \boldsymbol{x}_o^k) = \dfrac{p(\boldsymbol{x}_o^k \mid \boldsymbol{\theta})}{p^k(\boldsymbol{x}_o^k)} p^{k-1}(\boldsymbol{\theta} \mid \boldsymbol{x}_o^{k-1})$
7: **end for**
8: **Output:** Overconfident posterior distribution of $\boldsymbol{\theta}$.

---

**Algorithm 4** Overconfident posterior distribution for $\phi_2$ using NPE-PFN

1: **Input:** data vector, $x_o$; number of iterations, $K$; probability model, $p(x \mid \lambda)$; prior distribution, $p(\lambda \mid \phi_1)\, p(\phi_1 \mid \phi_2)\, p(\phi_2)$; number of training samples, $n_{tr}$
2: $\boldsymbol{x}_o^1 \leftarrow \boldsymbol{x}_o$ {Create a copy of $\boldsymbol{x}_o$.}
3: $\phi_{2,tr}^1 \sim^{n_{tr}} p(\phi_2)$ {Draw $n_{tr}$ samples of $\phi_2$ from the hyper-prior $p(\phi_2)$.}
4: $\phi_{1,tr}^1 \sim^{n_{tr}} p(\phi_1 \mid \phi_{2,tr}^1)$ {Draw $n_{tr}$ samples of $\phi_1$ conditional on the values of $\phi_{2,tr}$ generated in the previous step.}
5: $\lambda_{tr}^1 \sim^{n_{tr}} p(\lambda \mid \phi_{1,tr}^1)$ {Draw $n_{tr}$ samples of $\lambda$ conditional on the values of $\phi_{1,tr}$ generated in the previous step.}
6: $x_{tr}^1 \sim^{n_{tr}} p(x \mid \lambda_{tr}^1)$ {Draw $n_{tr}$ samples of $x$ conditional on the values of $\lambda_{tr}$ generated in the previous step.}
7: $\phi_{2,ts}^1 \sim^{n_{tr}} q(\phi_{2,ts} \mid \{x_o^1\}, \{\phi_{2,tr}^1, x_{tr}^1\})$ {Draw $n_{tr}$ samples from the posterior distribution of $\phi_2$ using a PFN model trained on $\{\phi_{2,tr}, x_{tr}\}$ and queried on $x_o$.}
8: **for** $k \in 2 : K$ **do**
9: $\quad \boldsymbol{x}_o^k \leftarrow \boldsymbol{x}_o$ {Create a copy of $\boldsymbol{x}_o$.}
10: $\quad \phi_{2,tr}^k \leftarrow \phi_{2,ts}^{k-1}$ {Use the samples from the posterior distr. of $\phi_2$ on iteration $k-1$ as the prior distr. samples in iteration $k$.}
11: $\quad \phi_{1,tr}^k \sim^{n_{tr}} p(\phi_1 \mid \phi_{2,tr}^k)$ {Draw $n_{tr}$ samples of $\phi_1$ conditional on the values of $\phi_{2,tr}$ generated in the previous step.}
12: $\quad \lambda_{tr}^k \sim^{n_{tr}} p(\lambda \mid \phi_{1,tr}^k)$ {Draw $n_{tr}$ samples of $\lambda$ conditional on the values of $\phi_{1,tr}$ generated in the previous step.}
13: $\quad x_{tr}^k \sim^{n_{tr}} p(x \mid \lambda_{tr}^k)$ {Draw $n_{tr}$ samples of $x$ conditional on the values of $\lambda_{tr}$ generated in the previous step.}
14: $\quad \phi_{2,ts}^k \sim^{n_{tr}} q(\phi_{2,ts} \mid \{x_o^k\}, \{\phi_{2,tr}^k, x_{tr}^k\})$ {Draw $n_{tr}$ samples from the posterior distribution of $\phi$ using a PFN model trained on $\{\phi_{2,tr}, x_{tr}\}$ and queried on $x_o$.}
15: **end for**
16: **Output:** Overconfident posterior distribution of $\phi_2$.

---

Note, however, that Algorithm 3 (and Algorithm 4) generates an over-confident asymptotic posterior distribution (i.e., a highly concentrated distribution that strongly underestimates the true uncertainty of any posterior inferences) because at each iteration it uses a copy of the original data, $\boldsymbol{x}_o$, as if it was an independent dataset drawn from the true data generating process.

It follows from standard Bayesian asymptotic theory (reviewed in Appendix E.2 above) that, as the sample size $n$ increases, the likelihood dominates the prior and the posterior distribution converges to a normal distribution centered at the maximum likelihood estimate, which converges to a degenerate point mass distribution around the true (or pseudo-true) parameter value as $n \rightarrow \infty$. But in the case of the overconfident posterior distribution the posterior is unable to converge to the true parameter value since the extra copies of the data do not provide any additional information. Rather, as shown in the next two results, the overconfident posterior collapses around the MLE of the original data.

**Theorem 4.** *Consider a dataset $\boldsymbol{w}_o = \{\boldsymbol{x}_o, \ldots, \boldsymbol{x}_o\}$ obtained by concatenating $K$ copies of $\boldsymbol{x}_o$. Let $\hat{\boldsymbol{\theta}}_o = \operatorname{argmax}_{\boldsymbol{\theta}} \log p(\boldsymbol{x}_o \mid \boldsymbol{\theta})$ and $\hat{\boldsymbol{\theta}}_{nK} = \operatorname{argmax}_{\boldsymbol{\theta}} \log p(\boldsymbol{w}_o \mid \boldsymbol{\theta})$ represent the maximum likelihood estimates of $\boldsymbol{\theta}$ computed on the $\boldsymbol{x}_o$ and $\boldsymbol{w}_o$ datasets, respectively. Then, under the modeling assumption that the copies of $\boldsymbol{x}_o$ composing $\boldsymbol{w}_o$ are treated as additional conditionally independent datasets given $\boldsymbol{\theta}$, it follows that $\hat{\boldsymbol{\theta}}_{nK} = \hat{\boldsymbol{\theta}}_o$ for all values of $K$.*

By combining Theorem 4 with the asymptotic results for the posterior distribution it follows that:

**Theorem 5.** *As $K \to \infty$, the over-confident posterior distribution of $\boldsymbol{\theta}$ generated by Algorithm 3 converges (under regularity conditions) to a degenerate point mass distribution centered at $\hat{\boldsymbol{\theta}}_o$.*

---

**Algorithm 5** NPE-PFN-based empirical Bayes for the two-level hierarchical model

1: **Input:** data vector, $x_o$; number of iterations, $K$; probability model, $p(x \mid \lambda)$; prior distribution, $p(\lambda \mid \phi_1) \, p(\phi_1 \mid \phi_2) \, p(\phi_2)$; number of training samples, $n_{tr}$; number of posterior samples, $n_{ts}$
2: $\phi_{2,ts}^K \leftarrow \text{OverconfidentPosterior}()$ {Call Algorithm 4 to generate the overconfident posterior distribution of $\phi_2$.}
3: $\phi_{2,tr} \leftarrow \phi_{2,ts}^K$ {Assign the samples drawn from the overconfident posterior as the prior for $\phi_2$.}
4: $\phi_{1,tr} \sim^{n_{tr}} p(\phi_1 \mid \phi_{2,tr}^1)$ {Draw $n_{tr}$ samples of $\phi_1$ conditional on the values of $\phi_{2,tr}$ generated in the previous step.}
5: $\lambda_{tr} \sim^{n_{tr}} p(\lambda \mid \phi_{1,tr})$ {Draw $n_{tr}$ samples of $\lambda$ conditional on the values of $\phi_{1,tr}$ generated in the previous step.}
6: $x_{tr} \sim^{n_{tr}} p(x \mid \lambda_{tr})$ {Draw $n_{tr}$ samples of $x$ conditional on the values of $\lambda_{tr}$ generated in the previous step.}
7: $\lambda_{ts} \sim^{n_{ts}} q(\lambda_{ts} \mid \{x_o\}, \{\lambda_{tr}, x_{tr}\})$ {Draw $n_{ts}$ samples from the posterior distribution of $\lambda$ using a PFN model trained on $\{\lambda_{tr}, x_{tr}\}$ and queried on $x_o$.}
8: $\phi_{1,ts} \sim^{n_{ts}} q(\phi_{1,ts} \mid \{\lambda_{ts}, x_o\}, \{\phi_{1,tr}, \lambda_{tr}, x_{tr}\})$ {Draw $n_{ts}$ samples from the posterior distribution of $\phi_1$ using a PFN model trained on $\{\phi_{1,tr}, \lambda_{tr}, x_{tr}\}$ and queried on $\{\theta_{ts}, x_o\}$.}
9: $\phi_{2,ts} \sim^{n_{ts}} q_\psi(\phi_2 \mid \{\phi_{1,ts}, \lambda_{ts}, x_o\}, \{\phi_{2,tr}, \phi_{1,tr}, \lambda_{tr}, x_{tr}\})$ {Draw $n_{ts}$ samples from the posterior distribution of $\phi_2$ using a PFN model trained on $\{\phi_{2,tr}, \phi_{1,tr}, \lambda_{tr}, x_{tr}\}$ and queried on $\{\phi_{1,ts}, \lambda_{ts}, x_o\}$.}
10: **Output:** $n_{ts}$ samples $\lambda_{ts}$, $\phi_{1,ts}$, and $\phi_{2,ts}$ from the posterior distribution of $\lambda$, $\phi_1$ and $\phi_2$.

---

## E.4. Proofs of Theorems 4 and 5

### E.4.1. PROOF OF THEOREM 4

*Proof.* Assume that the original dataset $\boldsymbol{x}_o$ contain $n$ examples, and let $\boldsymbol{w}_o = \{\boldsymbol{x}_o, \ldots, \boldsymbol{x}_o\}$ represent a larger dataset obtained by concatenating $K$ copies of $\boldsymbol{x}_o$.

Under the modeling assumption that the copies of $\boldsymbol{x}_o$ composing $\boldsymbol{w}_o$ are treated as additional conditionally independent datasets given $\boldsymbol{\theta}$ it follows that

$$\log p(\boldsymbol{w}_o \mid \boldsymbol{\theta}) = \log \prod_{k=1}^{K} p(\boldsymbol{x}_o \mid \boldsymbol{\theta}) = K \log p(\boldsymbol{x}_o \mid \boldsymbol{\theta}). \tag{22}$$

Hence,

$$\hat{\boldsymbol{\theta}}_{nK} = \operatorname*{argmax}_{\boldsymbol{\theta}} \log p(\boldsymbol{w} \mid \boldsymbol{\theta}) = \operatorname*{argmax}_{\boldsymbol{\theta}} K \log p(\boldsymbol{x}_o \mid \boldsymbol{\theta}) = \operatorname*{argmax}_{\boldsymbol{\theta}} \log p(\boldsymbol{x}_o \mid \boldsymbol{\theta}) = \hat{\boldsymbol{\theta}}_o , \tag{23}$$

since multiplying by $K$ does not change the maximizer.

Note that even though the copies of $\boldsymbol{x}_o$ are not truly independent conditional on $\boldsymbol{\theta}$, standard MLE software will still handle the combined dataset as if the copies were conditionally independent (i.e., it assumes that the likelihood factorizes across all rows of the combined dataset). $\qquad\square$

### E.4.2. PROOF OF THEOREM 5

*Proof.* Assume that the original dataset $\boldsymbol{x}_o$ contain $n$ examples. The application of Algorithm 3 with $\boldsymbol{x}^k = \boldsymbol{x}_o$, for $k = 1, \ldots, K$, generates the over-confident posterior distribution $p(\theta \mid \boldsymbol{w}_o)$, where $\boldsymbol{w}_o = \{\boldsymbol{x}_o, \ldots, \boldsymbol{x}_o\}$ corresponds to the larger dataset obtained by concatenating $K$ copies of $\boldsymbol{x}_o$. (This distribution is over-confident because Algorithm 3 treats the dataset copies as if they were independently sampled from the true data generation process. In other words, it assumes the the sample size in $nK$ while, in reality, the effective sample size of is $n$)

By increasing the number of iterations $K$ (and, therefore, $nK$), it follows from Result 3 that the over-confident posterior distribution converges to a normal distribution centered around $\hat{\theta}_{nK}$ (the MLE computed on the dataset $w_o$) and with variance inversely proportional to $nK$. From Theorem 4 we have that $\hat{\theta}_{nK} = \hat{\theta}_o$ for all values of $K$. Hence, as $K$ increases to infinity the posterior generated by Algorithm 3 converges to a degenerate point mass distribution centered at $\hat{\theta}_o$, the MLE computed on the original data $x_o$. $\qquad\square$

## F. Extended description for the MAP/MLE equivalence experiments

### F.1. Experimental setup

For the MAP/MLE equivalence experiments we evaluated 9 distinct simulation settings encompassing all combinations of sample sizes, $n = \{30, 60, 120\}$, and number of in-context examples used to train the NPE-PFN model, $n_{tr} = \{250, 500, 1000\}$, as described in Table 2.

*Table 2.* Simulation settings evaluated in the MAP/MLE equivalence experiments. $n$ represents the observed data sample size. $n_{tr}$ represents the number of in-context examples used to train the NPE-PFN model.

| SIMULATION SETTING: | 1 | 2 | 3 | 4 | 5 | 6 | 7 | 8 | 9 |
|---|---|---|---|---|---|---|---|---|---|
| SAMPLE SIZE ($n$): | 30 | 60 | 120 | 30 | 60 | 120 | 30 | 60 | 120 |
| NUMBER OF TRAINING EXAMPLES ($n_{tr}$): | 250 | 250 | 250 | 500 | 500 | 500 | 1000 | 1000 | 1000 |

For each probability model in each experimental setting, we performed 30 simulations based on distinct true $\lambda$ values where:

- $\lambda$ was randomly sampled from a $U(0.1, 0.9)$ distribution for the the bernoulli and negative binomial models (for which $\lambda$ can only assume values in the interval $[0, 1]$.

- $\lambda$ was randomly sampled from a $U(0.1, 3)$ distribution for the poisson, exponential, and normal model with known mean (for which $\lambda \in \mathbb{R}^+$).

- $\lambda$ was randomly sampled from a $U(-3, 3)$ distribution for the normal model with known variance (for which $\lambda \in \mathbb{R}$).

For each model we draw $n$ samples from the probability model and computed 1000 draws from the posterior distribution using $n_{tr}$ in-context learning examples during NPE-PFN training. In our evaluations, we adopted uniform-half-t priors with scale parameter $s_\lambda = 10$ and degrees of freedom parameter $d_\lambda = 3$.

### F.2. Extended results

Section 3 in the main text report results for the MAP/MLE equivalence evaluations pooled together across the 9 distinct simulation settings described in Table 2. Here, we report results for each simulation setting separately (as well as additional evaluations).

Because the NPE-PFN approach produces approximate posterior distributions, it is important to evaluate how well the theoretical results in Theorems 1 and 2 hold in practice. To this end, we adopt the relative absolute distance (RAD) metric, defined as,

$$\text{RAD} = \frac{|\hat{\lambda}_{ML} - \hat{\lambda}_{MODE}|}{\text{PostRange}}, \qquad (24)$$

where $\hat{\lambda}_{ML}$ corresponds to the MLE estimate of $\lambda$, $\hat{\lambda}_{MODE}$ corresponds to the mode of the approximate posterior $p(\lambda \mid x_o)$ generated by the NPE-PFN methodology, and PostRange corresponds to the range of the approximate posterior distribution. Note that it is necessary to divide the absolute distance $|\hat{\lambda}_{ML} - \hat{\lambda}_{MODE}|$ by the posterior range because the $\lambda$ parameters can have different scales across models and across simulations. By dividing the absolute distance by the range, the RAD metric measures the relative difference between the MLE estimate and the posterior mode in terms of a percentage of the range of the posterior distribution. For instance, a RAD score equal to 0.01 means that the (absolute) distance between the posterior mode and the MLE estimate represents only 1% of the total range of the posterior.

Tables 3 to 11 report the mean RAD scores (alongside the standard deviations) for each one of the simulation settings. (The means and standard deviations were computed across the 30 simulations of each experiment based on different $\lambda$ values.)

The model names were abbreviated as follows: poisson (PO), exponential (EX), normal with known variance (NM), normal with known mean (NV), bernoulli (BE), and negative binomial (NB).

*Table 3.* Simulation setting 1: $n = 30$, $n_{tr} = 250$.

| MODEL | P0 | EX | NM | NV | BE | NB |
|---|---|---|---|---|---|---|
| MEAN | 0.031 | 0.037 | 0.016 | 0.029 | 0.016 | 0.036 |
| SD | 0.022 | 0.029 | 0.010 | 0.026 | 0.011 | 0.026 |

*Table 4.* Simulation setting 1: $n = 60$, $n_{tr} = 250$.

| MODEL | P0 | EX | NM | NV | BE | NB |
|---|---|---|---|---|---|---|
| MEAN | 0.032 | 0.046 | 0.023 | 0.028 | 0.016 | 0.037 |
| SD | 0.020 | 0.033 | 0.014 | 0.024 | 0.011 | 0.029 |

*Table 5.* Simulation setting 3: $n = 120$, $n_{tr} = 250$.

| MODEL | P0 | EX | NM | NV | BE | NB |
|---|---|---|---|---|---|---|
| MEAN | 0.028 | 0.060 | 0.022 | 0.035 | 0.015 | 0.036 |
| SD | 0.021 | 0.042 | 0.014 | 0.029 | 0.009 | 0.027 |

*Table 6.* Simulation setting 4: $n = 30$, $n_{tr} = 500$.

| MODEL | P0 | EX | NM | NV | BE | NB |
|---|---|---|---|---|---|---|
| MEAN | 0.023 | 0.034 | 0.016 | 0.035 | 0.015 | 0.029 |
| SD | 0.018 | 0.025 | 0.011 | 0.028 | 0.014 | 0.020 |

*Table 7.* Simulation setting 5: $n = 60$, $n_{tr} = 500$.

| MODEL | P0 | EX | NM | NV | BE | NB |
|---|---|---|---|---|---|---|
| MEAN | 0.027 | 0.041 | 0.015 | 0.032 | 0.015 | 0.030 |
| SD | 0.017 | 0.029 | 0.014 | 0.027 | 0.011 | 0.027 |

*Table 8.* Simulation setting 6: $n = 120$, $n_{tr} = 500$.

| MODEL | P0 | EX | NM | NV | BE | NB |
|---|---|---|---|---|---|---|
| MEAN | 0.024 | 0.059 | 0.019 | 0.042 | 0.015 | 0.026 |
| SD | 0.020 | 0.038 | 0.016 | 0.031 | 0.012 | 0.028 |

*Table 9.* Simulation setting 7: $n = 30$, $n_{tr} = 1000$.

| MODEL | P0 | EX | NM | NV | BE | NB |
|---|---|---|---|---|---|---|
| MEAN | 0.020 | 0.028 | 0.013 | 0.030 | 0.015 | 0.027 |
| SD | 0.017 | 0.025 | 0.012 | 0.025 | 0.010 | 0.018 |

*Table 10.* Simulation setting 8: $n = 60$, $n_{tr} = 1000$.

| MODEL | P0 | EX | NM | NV | BE | NB |
|---|---|---|---|---|---|---|
| MEAN | 0.017 | 0.041 | 0.016 | 0.036 | 0.014 | 0.028 |
| SD | 0.014 | 0.033 | 0.012 | 0.031 | 0.008 | 0.022 |

*Table 11.* Simulation setting 9: $n = 120$, $n_{tr} = 1000$.

| MODEL | P0 | EX | NM | NV | BE | NB |
|---|---|---|---|---|---|---|
| MEAN | 0.022 | 0.060 | 0.015 | 0.044 | 0.016 | 0.032 |
| SD | 0.017 | 0.045 | 0.016 | 0.029 | 0.012 | 0.024 |

In addition to computing the relative absolute distance between the MLE and posterior mode estimates, we have also compared the mean squared error (MSE) between the true $\lambda$ values and the MLE estimates,

$$MSE_{ML} = \frac{1}{B} \sum_{i=1}^{B} (\lambda_i - \hat{\lambda}_{ML,i})^2 \,, \tag{25}$$

and the MSE between the true $\lambda$ and the posterior modes,

$$MSE_{MODE} = \frac{1}{B} \sum_{i=1}^{B} (\lambda_i - \hat{\lambda}_{MODE,i})^2 \,. \tag{26}$$

Figure 2 reports the results, and shows that the $MSE_{MODE}$ scores (blue boxplots) tends to track the $MSE_{ML}$ scores (red boxplots) across all models and simulation settings. As expected, the MSE scores tended to be lower for the simulation settings involving larger sample sizes.

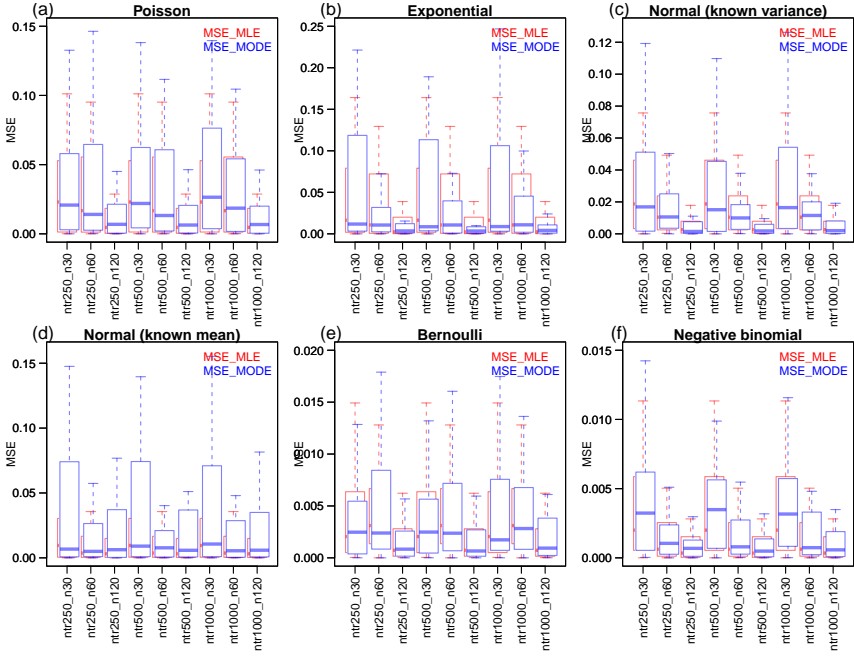

*Figure 2.* Comparison of mean squared errors computed using the MLE estimate (red boxplots) versus the estimated mode of the approximate posterior distribution generated using NPE-PFN methodology (blue boxplots).

## G. Performance comparison between uniform-halt-t and reference priors

Here we compare the performance of the proposed uniform-half-t prior against reference priors for the poisson, exponential, normal with known variance, and normal with known mean models.

### G.1. Reference priors

Reference priors are a class of objective Bayesian priors constructed by maximizing the expected Kullback-Leibler divergence between the prior and the posterior, thereby selecting the prior that allows the data to have the greatest possible influence on inference. See Bernardo and Smith (1994) for further details.

For each of the 4 probability models evaluated with uniform-half-t priors in our experiments, we list below their probability (mass) density function, $p(x \mid \lambda)$, their reference prior, $\pi(\lambda)$, and their reference posterior, $\pi(\lambda \mid x)$.

Poisson model:

$$p(x_i \mid \lambda) = Poisson(x_i \mid \lambda)\,, \quad \pi(\lambda) \propto \lambda^{-1/2}\,, \quad \pi(\lambda \mid \boldsymbol{x}) = Gamma\Big(\lambda \mid \frac{1}{2} + \sum_{i=1}^{n} x_i\,,\, n\Big)\,. \tag{27}$$

Exponential model:

$$p(x_i \mid \lambda) = Exponential(x_i \mid \lambda)\,, \quad \pi(\lambda) \propto \lambda^{-1}\,, \quad \pi(\lambda \mid \boldsymbol{x}) = Gamma\Big(\lambda \mid n\,,\, \sum_{i=1}^{n} x_i\Big)\,. \tag{28}$$

Normal model (with known variance $\sigma^2$):

$$p(x_i \mid \lambda) = Normal(x_i \mid \lambda, \sigma^2)\,, \quad \pi(\lambda) \propto 1\,, \quad \pi(\lambda \mid \boldsymbol{x}) = Normal\Big(\lambda \mid \bar{x}\,,\, \frac{\sigma^2}{n}\Big)\,. \tag{29}$$

Normal model (with known mean $\mu$):

$$p(x_i \mid \lambda) = Normal(x_i \mid \mu, \lambda^2)\,, \quad \pi(\lambda) \propto \lambda^{-2}\,, \quad \pi(\lambda \mid \boldsymbol{x}) = InverseGamma\Big(\lambda \mid \frac{n}{2}\,,\, \frac{\sum_{i=1}^{n}(x_i - \mu)^2}{2}\Big)\,. \tag{30}$$

Note that while the reference Bayesian analysis methodology does not aim directly at generating inferences equivalent to MLE based inferences, in practice, the posterior means of reference posterior distributions closely approximate or exactly match the respective sampling model MLE. For instance, the means of the reference posterior distributions for the exponential and normal (with known variance) models are given, respectively, by $1/\bar{x}$ and $\bar{x}$ and match exactly the MLE for these models. The mean of the reference posterior for the normal model (with known mean) is given by $\sum_{i=1}^{n}(x_i - \mu)^2/(n-2)$ whereas the MLE is given by $\sum_{i=1}^{n}(x_i - \mu)^2/n$. The mean of the reference posterior for the poisson model is given by $\bar{x} + 1/(2n)$ whereas the MLE is given by $\bar{x}$.

### G.2. Experimental results

For each model, we compared the mean squared error (MSE) of the posterior means obtained using the uniform-half-t prior with those obtained using the corresponding model-specific reference prior, described in Appendix G.1.

For each probability model we performed 100 simulations based on distinct true $\lambda$ values randomly sampled from a $U(0.1, 3)$ distribution for models where $\lambda \in \mathbb{R}^+$ (namely, the poisson, exponential, and normal model with known mean) and sampled from a $U(-3, 3)$ distribution for the normal model with known variance. For each model we draw $n = 15$ samples from the probability model and computed the respective posterior distributions based on the reference and uniform-half-t priors with scale parameter $s_\lambda = 10$ and degrees of freedom parameter $d_\lambda = 3$. We generated 1000 draws from each posterior distribution. The NPE-PFN computations were based on training datasets containing 1000 examples.

Because the reference priors for these models are improper and the NPE-PFN framework requires sampling from the prior, direct prior sampling is not possible. Instead, we generated data by sampling directly from the corresponding reference posterior distributions, which admit closed-form expressions for all four models (described in Appendix G.1).

Figure 3 report the results. For all probability models, the MSE scores obtained by the posterior distributions generated via NPE-PFN with uniform-half-t priors (blue boxplots) closely approximate the MSE scores from the reference posteriors (red boxplots), which represent the state-of-the-art method for objective Bayesian analysis. (Note that in these additional experiments, the MSE scores are computed comparing the true $\lambda$ value against the mean of the posterior distributions.)

These results illustrate how the NPE-PFN approach fitted with the exact same uniform-half-t default prior closely track the results generated by the reference priors, which employs model specific priors. As expected, these results illustrate that, in practice, our proposed approach generates posterior inferences that closely match the posterior inferences obtained by this widely used objective Bayesian approach. (Note that the goal is to show that our approach generates similar inferences to reference Bayesian analysis in a more automatic way, and not that it produces more accurate parameter estimates.)

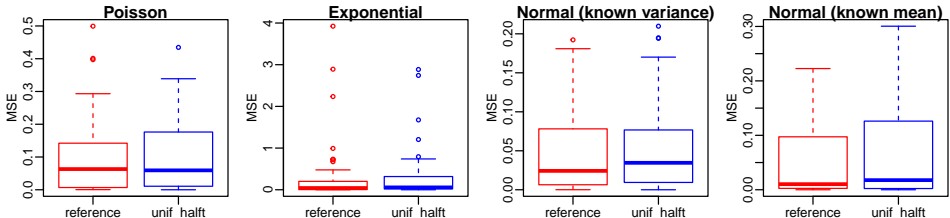

*Figure 3.* Posterior MSE comparisons based on reference priors (red) vs uniform-half-t priors (blue).

## H. Extended description for the NPE-PFN-based EB experiments

### H.1. Model description

The current state-of-the-art solution for the hierarchical normal means model described in section 4 is to adopt weakly informative priors from t-family distributions for the $\mu$, $\sigma$, and $\tau$ parameters. For our evaluations we adopt half-t distributions for the variance component parameters $\sigma$ and $\tau$, and a t-distribution for $\mu$. The corresponding full Bayesian hierarchical model is given by,

$$p(x_{ij} \mid \mu, \alpha_j, \sigma) \sim N(\mu + \alpha_j , \sigma^2), \quad i = 1, \ldots, n_j, \ j = 1, \ldots, J, \tag{31}$$

$$p(\alpha_j \mid \tau) \sim N(0 , \tau^2), \quad j = 1, \ldots, J, \tag{32}$$

$$p(\mu) \sim t_{d_\mu}(m_\mu , s_\mu^2), \tag{33}$$

$$p(\sigma) \sim ht_{d_\sigma}(s_\sigma^2), \tag{34}$$

$$p(\tau) \sim ht_{d_\tau}(s_\tau^2), \tag{35}$$

and can be implemented via MCMC using the `brms` R package.

Our proposed two-level hierarchical model with a hierarchical uniform-half-t prior is given by,

$$p(x_{ij} \mid \mu, \alpha_j, \sigma) \sim N(\mu + \alpha_j , \sigma^2), \quad i = 1, \ldots, n_j, \ j = 1, \ldots, J, \tag{36}$$

$$p(\alpha_j \mid \tau) \sim N(0 , \tau^2), \quad j = 1, \ldots, J, \tag{37}$$

$$p(\mu \mid \gamma) \sim U(-\gamma, \gamma), \tag{38}$$

$$p(\sigma \mid \eta) \sim U(0, \eta), \tag{39}$$

$$p(\tau \mid \delta) \sim U(0, \delta), \tag{40}$$

$$p(\gamma) \sim ht_{d_\gamma}(s_\gamma^2), \tag{41}$$

$$p(\eta) \sim ht_{d_\eta}(s_\eta^2), \tag{42}$$

$$p(\delta) \sim ht_{d_\delta}(s_\delta^2). \tag{43}$$

In our experiments we evaluate this model in two ways. First, as a standard two-level Bayesian hierarchical model, where we generate from the posterior distribution,

$$p^{2H}(\mu, \sigma, \tau, \alpha_1, \ldots, \alpha_J \mid x_o) \propto \tag{44}$$
$$\Big[ \prod_{i=1}^{n_j} \prod_{j=1}^{J} p(x_{ij} \mid \mu, \alpha_j, \sigma) p(\alpha_j \mid \tau) \Big] p(\mu \mid \gamma) \, p(\sigma \mid \eta) \, p(\tau \mid \delta) \, p(\gamma) \, p(\eta) \, p(\delta),$$

using the NPE-PFN approach.

Second, using the empirical Bayes strategy, proposed in section 2.2.2, where we replacement of the priors $p(\gamma)$, $p(\eta)$, and $p(\delta)$ by the respective overconfident posterior distributions, $p(\gamma \mid \boldsymbol{w}_o)$, $p(\eta \mid \boldsymbol{w}_o)$, and $p(\delta \mid \boldsymbol{w}_o)$ (generated by Algorithm

4), and generate samples from the empirical Bayes posterior distribution,

$$p^{EB}(\mu,\sigma,\tau,\alpha_1,\ldots,\alpha_J \mid x_o) \propto \tag{45}$$

$$\Big[\prod_{i=1}^{n_j}\prod_{j=1}^{J} p(x_{ij} \mid \mu, \alpha_j, \sigma)p(\alpha_j \mid \tau)\Big]p(\mu \mid \gamma)\,p(\sigma \mid \eta)\,p(\tau \mid \delta)\,p(\gamma \mid \boldsymbol{w}_o)\,p(\eta \mid \boldsymbol{w}_o)\,p(\delta \mid \boldsymbol{w}_o)\,,$$

using the NPE-PFN approach (as described in Algorithm 5).

All the EB-NPE-PFN results reported in this paper were generated using the following parameter order:

$$\mu,\ \sigma,\ \alpha_1,\ \alpha_2,\ \alpha_3,\ \tau,\ \gamma,\ \omega,\ \eta,\ \text{and } \delta. \tag{46}$$

### H.2. Experimental setup

We performed 100 simulations based on distinct values of $\mu$, $\sigma$ and $\tau$ randomly sampled according to $\mu \sim U(0.1, 1)$, $\sigma \sim U(0.1, 1)$, and $\tau \sim U(0.1, 1)$. For each simulation we generated data from the hierarchical normal means model with $J = 3$ groups by randomly sampling random effects $\alpha_j \sim N(0, \tau^2)$, and drawing $n_j = 10$ samples from each group from the probability model $x_{ij} \sim N(\mu + \alpha_j, \sigma^2)$. The hierarchical model based on weakly informative priors described in equations 31 to 35 was implemented via MCMC using the brms R package, with $m_\mu$ set to 0, scale parameters set to $s_\mu = s_\sigma = s_\tau = 10$ and degrees of freedom parameters set to $d_\mu = d_\sigma = d_\tau = 3$. The posterior was generated using 4 separate chains, and we discarded the first 500 samples as burn-in.

The two-level hierarchical model based on uniform-half-t priors described in equations 36 to 43 was implemented using the NPE-PFN strategy, based on training datasets containing 1000 examples. We adopted scale parameter $s_\gamma = s_\eta = s_\delta = 10$ and degrees of freedom parameter $d_\gamma = d_\eta = d_\delta = 3$ for the uniform-half-t priors. For each posterior distribution, we generated 1,000 samples.

The EB-NPE-PFN approach was implemented using the same two-stage hierarchical Bayesian model as in the NPE-PFN approach but where we replace the second stage priors by the respective overconfident posterior distributions (as described in equations 44 and 45). The overconfident posteriors were generated using $K = 10$ iterations.

### H.3. Additional results

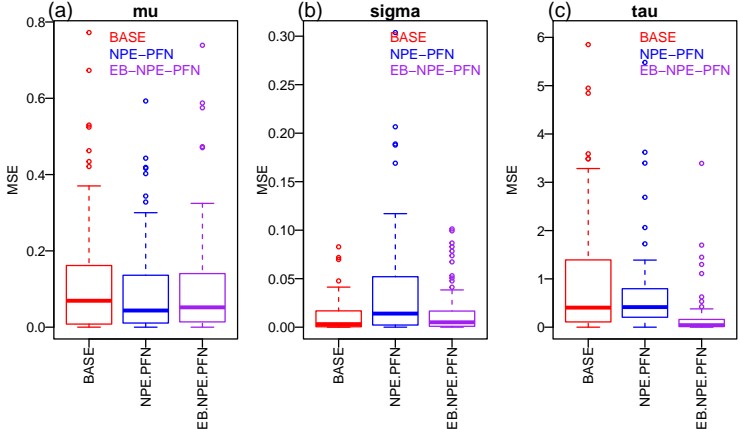

*Figure 4.* MSE comparisons between the BASE (red), NPE-PFN (blue), and EB-NPE-PFN (purple) approaches. The BASE model is described in equations 31 to 35 and implemented via MCMC with the `brms` package. The NPE-PFN approach is implemented with the two-stage hierarchical Bayesian model described in equations 36 to 43. The EB-NPE-PFN approach is implemented using the same two-stage hierarchical Bayesian model as in the NPE-PFN approach but where we replace the second stage priors by the respective overconfident posterior distributions. Because the NPE-PFN and EB-NPE-PFN are based on the same underlying hierarchical model, the difference in performance between these two approaches is explained by the EB approach. For the $\tau$ parameter (panel c) we observe that the EB-NPE-PFN approach provides a strong improvement in performance over the NPE-PFN (which is now comparable with BASE). For the $\mu$ parameter (panel a) the results where comparable for all methods. For the $\sigma$ parameter (panel b), while NPE-PFN shows worse performance than EB-NPE-PFN and BASE (the differences in performance are, nonetheless, small - note the smaller dynamic range in the y-axis of panel b compared to panels a and c).

# I. Limitations

Because our approach employs TabPFN as the underlying inferential engine, it inevitably inherits the limitations of this tabular foundation model. For the model version used in our experiments, namely, TabPFNv2 (Hollmann et al., 2025), these limitations include a soft limit of 10,000 rows and 500 columns for the datasets the model can handle. These limits, however, have already been greatly relaxed in a recently released new version (TabPFN-2.5) which extended the data size limits to 50,000 rows and 2,000 columns (Grinsztajn et al., 2025).

In any case, the column limit of 500 (or 2000) columns imposed by the TabPFN model limits the direct application of our approach to observed datasets with sample sizes larger than the column limit. This happens because in the statistical inference applications investigated in this paper, the training data that is fed into the TabPFN model is organized as a table with rows corresponding to the simulated training examples and columns corresponding to the concatenation of the model parameters sampled from the prior distributions and the data sampled from the probability model (conditional on the sampled parameter values). Hence, the number of columns of the training data table is $l + n$, where $l$ represents the number of parameters and $n$ represents the sample size of the observed data $\boldsymbol{x}_o$.

We point out, however, that while direct application of the TabPFN model is unfeasible when $l + n > 500$ (or $l + n > 2000$) it is still possible to run our Bayesian statistical analyses for larger datasets using the sequential updating property of Bayesian inference (described in Appendix E.1). The basic idea is to split the sampled data into multiple smaller chunks (which can be handled by the TabPFN model) and generate the posterior distribution in a sequential fashion where the posterior from one data chunk becomes the prior for the next data chunk. This approach can be easily implemented by modifying Algorithm 3 by changing line 5 to receive a new data chunk instead of a data copy at each iteration of the algorithm (generating, in this way, a bona fide posterior distribution instead of a overconfident one).

