# OpenReview forum: "Objective and data-driven Bayesian inference using TabPFN models"
_ICML.cc/2026/Workshop/FMSD — FMSD @ ICML 2026 Poster_

### Official Review · Reviewer_m9Md · 2026-05-20
**Fair contribution but limited empirical validation**

**Rating:** 6
**Confidence:** 3

**Review:**

This paper applies TabPFN for both objective and empirical Bayesian inference. The paper proceeds in two steps. First, it introduces a class of default priors with the property that the maximum a posteriori estimate coincides with the maximum likelihood estimate. The paper shows that this criterion leads to a simple hierarchical prior family that is model agnostic and easy to sample from, in contrast to model-specific reference priors. Second, the paper proposes a method to approximate maximum likelihood estimation of hyperparameters using "overconfident" posteriors. Then, it uses these approximate MLEs as empirical Bayes to improve inference.

The main strength of the paper lies in the derivation of the default prior family, which is simple and easy to manipulate. Additionally, the proposed overconfident posterior trick to estimate empirical Bayes is novel to me. It would be interesting to include a discussion on the failure mode of this trick (e.g., to what extent one can increase the number of copies $K$?). Overall, the two contributions are well-motivated and supported by theoretical insights.

First, I find the paper would benefit from clearer flow, structure, and illustrations. In its current form, it is hard to grasp the high-level details without referring to the appendices (and referenced papers). Second, while the technical motivation of the work is sound, I find the empirical validation is rather limited, restricted to a simple hierarchical model. It would be interesting to see how these results generalize to more realistic tasks. For example, since TFMs (TabPFN and TabICL) are trained using synthetic data generated from structural causal priors, would it not make sense to apply the proposed method to infer the parameters of causal graph models?

Overall, the core contribution of the paper is fair, but the empirical validation is limited.

---

### Official Review · Reviewer_orsm · 2026-05-21
**Original NPE-FPN extension with some clarification needed**

**Rating:** 6
**Confidence:** 4

**Review:**

This paper extends Neural Posterior Estimation with Prior-fitted networks (NPE-PFN) to the tasks of objective Bayesian inference and empirical Bayes. It introduces two methodological ideas: 1) a hierarchical unfiorm-half-t prior to achieve sampleable distribtutions where MAP and MLE are equivalent over unbounded parameters spaces. 2) an empricial Bayes framework that computes the hyperparameters MLE based on iteratively extracting overconfident predictions from tabPFN.

Strengths:

1. Finding a way to perform automated statistics without needing analytical reference prior derivations or heavy optimization loops is highly valuable for the tabular foundatoin model community.
2. The experiments demonstrate an empirical advantage over standard MCMC.

Areas for improvement

1. I did not go over the proof of Theorem 1 rigorously but I don't see how this method can provide a prior that does not decay for unbounded parameters. Therefore I don't not see how the claim that MLE and MAP are equivalent hold. I think this should be either toned down or better described.
2. The experimental scenarios are quite narrow. This is fine for a workshop paper but should be extended in further submission, showing a larger range of parameter values and also contain multi-parameter setups.

Justification of Score

The paper outlines a creative and original method for leveraging the strengths of tabular foundation models to solve classical statistical estimation constraints.  While I am not sure about theorem 1 and the experiments rely on narrow toy and edge cases, the core framework provides a practical approach to perform new Bayesian inference tasks. Provided a clarifications regarding Thm 1, I think this work will provide a valuable addition to the workshop.